# Heart fossilization is possible and informs the evolution of cardiac outflow tract in vertebrates

Lara Maldanis[1,2†], Murilo Carvalho[2,3†], Mariana Ramos Almeida[4], Francisco Idalécio Freitas[5], José Artur Ferreira Gomes de Andrade[6], Rafael Silva Nunes[7], Carlos Eduardo Rochitte[8], Ronei Jesus Poppi[4], Raul Oliveira Freitas[7], Fábio Rodrigues[9], Sandra Siljeström[10], Frederico Alves Lima[7], Douglas Galante[7], Ismar S Carvalho[11], Carlos Alberto Perez[7], Marcelo Rodrigues de Carvalho[3], Jefferson Bettini[12], Vincent Fernandez[13]*, José Xavier-Neto[2]*

[1]Department of Pharmacology, University of Campinas, Campinas, Brazil; [2]Brazilian Biosciences National Laboratory, Campinas, Brazil; [3]Department of Zoology, Biosciences Institute, University of São Paulo, São Paulo, Brazil; [4]Institute of Chemistry, University of Campinas, Campinas, Brazil; [5]Geopark Araripe, Crato, Brazil; [6]National Department of Mineral Production, Ministry of Mines and Energy, Crato, Brazil; [7]Brazilian Synchrotron Light Laboratory, Campinas, Brazil; [8]Heart Institute, InCor, University of São Paulo, São Paulo, Brazil; [9]Institute of Chemistry, University of São Paulo, São Paulo, Brazil; [10]Department of Chemistry, Materials, and Surfaces, SP Technical Research Institute of Sweden, Borås, Sweden; [11]Departamento de Geologia, Universidade Federal do Rio de Janeiro, Rio de Janeiro, Brazil; [12]Brazilian Nanotechnology National Laboratory, Campinas, Brazil; [13]European Synchrotron Radiation Facility, Grenoble, France

*For correspondence: vincent. fernandez@esrf.fr (VF); xavier. neto@lnbio.cnpem.br (JXN)

†These authors contributed equally to this work

Competing interests: The authors declare that no competing interests exist.

**Abstract** Elucidating cardiac evolution has been frustrated by lack of fossils. One celebrated enigma in cardiac evolution involves the transition from a cardiac outflow tract dominated by a multi-valved conus arteriosus in basal actinopterygians, to an outflow tract commanded by the non-valved, elastic, bulbus arteriosus in higher actinopterygians. We demonstrate that cardiac preservation is possible in the extinct fish *Rhacolepis buccalis* from the Brazilian Cretaceous. Using X-ray synchrotron microtomography, we show that *Rhacolepis* fossils display hearts with a conus arteriosus containing at least five valve rows. This represents a transitional morphology between the primitive, multivalvar, conal condition and the derived, monovalvar, bulbar state of the outflow tract in modern actinopterygians. Our data rescue a long-lost cardiac phenotype (119-113 Ma) and suggest that outflow tract simplification in actinopterygians is compatible with a gradual, rather than a drastic saltation event. Overall, our results demonstrate the feasibility of studying cardiac evolution in fossils.

## Introduction

The hearts of ray-finned fishes (actinopterygians) are presently described as a succession of four muscular chambers that perform inflow (sinus venosus and atrium) and outflow (ventricle and conus arteriosus) roles, followed by the bulbus arteriosus, a terminal, non-chambered, elastic cardiac segment (*Simões-Costa et al., 2005*; *Grimes et al., 2006*; *Durán et al., 2008*).

**eLife digest** Modern research has majorly advanced our understanding of how the heart works, and has led to new therapies for cardiac diseases. However, little is known about how the heart has evolved throughout the history of animals with backbones – a group that is collectively referred to as vertebrates. This is partly because the heart is made from soft muscle tissue, which does not fossilize as often as harder tissues such as bones.

Even though fossils of soft tissues are rare, paleontologists have already unearthed fossils of other soft organs such as the stomach and umbilical cord. These discoveries suggested that there was hope of finding fossil hearts, and now Maldanis, Carvalho et al. have indeed discovered fossil hearts in two specimens of an extinct species of bony fish called *Rhacolepis buccalis*. These fish were alive over 113 million years ago during the Cretaceous period, in an area that is now modern-day Brazil.

Like all known vertebrates, these *R. buccalis* fossils have valves between the heart and the major artery that carries blood out of the heart. Such valves are vital because they prevent pumped blood from flowing back into the heart. However, oddly, *R. buccalis* fossils show five of these valves, which is more than any advanced bony fish that is alive today. Comparing this with the situation in other fish species suggests that vertebrate hearts gradually evolved to become progressively simpler.

This discovery shows that it is possible to study heart evolution with fossils. Maldanis, Carvalho et al. hope that their findings will stimulate researchers from all over the world to examine the fossils of well-preserved animals in search of clues to help reconstruct the major steps in the evolution of the vertebrate heart.

In basal actinopterygians, the conus arteriosus dominates the cardiac outflow, while in teleosts, it is the bulbus arteriosus that prevails, a notion that harks back to *Gegenbaur (1866)* and before. The conus arteriosus displays multiple fibrous valve rows, a character state that represents the general gnathostome condition, primitively retained in basal actinopterygian groups (*Durán et al., 2008*; *Boas, 1880*; *Boas, 1901*; *Schib et al., 2002*; *Xavier-Neto et al., 2010*; *Parsons, 1929*; *Icardo et al., 2002a*; *Durán et al., 2014*; *Icardo et al., 2002b*). The multiple conal valve rows of basal actinopterygians prevent backflow and protect the delicate gill vessels from the elevated pulsations generated by the ventricle (*Satchell and Jones, 1967*). In contrast, in derived actinopterygians such as the teleost zebrafish, the valveless bulbus arteriosus protects the gills through its prominent elastic properties (i.e. functioning as a windkessel [*Farrell, 1979*]). Thus, teleost hearts display only one valve row at the bulbo-ventricular transition, which is now regarded as an evolutionary remnant of the conus arteriosus (*Grimes et al., 2006*).

The transition from a heart packed with dozens of outflow tract valves in basal actinopterygians, such as in the genus *Polypterus* (*Durán et al., 2014*), to the single valve row in the cardiac outflow tract of derived actinopterygians, such as in the cypriniform teleost *Danio rerio* (the zebrafish) (*Grimes and Kirby, 2009*) represents a celebrated, hundred-year-old, case of secondary cardiac simplification. The emphasis on the bulbus arteriosus, rather than on the conus arteriosus in teleosts, and the concurrent reduction in the number of outflow valve rows are presently almost completely unconstrained in evolutionary and developmental times. We know that the primitive actinopterygian *Polypterus* diverged from other actinopterygian lineages (including the zebrafish) by about 390 Mya (*Takeuchi et al., 2009*) and that the elastic bulbus arteriosus of teleosts represents a very late ontogenetic addition, being added to the heart only after cardiac chambers are formed (*Grimes et al., 2006*; *Grimes and Kirby, 2009*). With such limited information, it is impossible to answer whether outflow tract simplification in teleosts represented another case of phyletic gradualism, or resulted from drastic developmental effects. Significant morphological changes are sometimes associated with major genetic changes, such as large-scale gene duplications and/or changes in the function of genes with major developmental effects, both known to have taken place in teleost evolution (*Shapiro et al., 2004*; *Shubin et al., 1997*). Knowledge of morphological transitions between character states is critical to the construction of any evolutionary hypothesis. Thus, the first steps toward

the understanding of any evolutionary modification ideally involve the discovery of intermediate morphologies.

Unfortunately, there are no universally recognized descriptions of fossilized vertebrate chambered hearts (*Xavier-Neto et al., 2010*; *Janvier, 1996*; *Rowe et al., 2001*; *Fisher et al., 2000*; *Cleland et al., 2011*). Although provocative clues accumulate (*Janvier, 1996*; *Rowe et al., 2001*; *Fisher et al., 2000*; *Cleland et al., 2011*; *Shu et al., 2003*; *Carvalho and Maisey, 1996*, *Janvier et al., 1991*), none of the specimens described so far retained enough original attributes to establish beyond dispute that cardiac preservation is possible. Part of the problem is that the heart is formed by soft tissues, which fossilize only under special conditions (*Martill, 1988*). However, other soft organs have been described in the Cretaceous of Araripe, Brazil (*Martill, 1988*; *1990*; *Pradel et al., 2009*; *Brito et al., 2010*) and even in Paleozoic fishes (380 million-years old) from the Gogo Formation in Australia (*Trinajstic et al., 2007*; *2013*; *Long et al., 2008*) and Antarctica (*Young et al., 2010*), which suggests that the difficulty lies not with cardiac preservation, but with the lack of a systematic search.

## Results

In the course of a wider search for fossil hearts, we fortuitously found evidence for a long and gradual evolutionary reduction of the conus arteriosus and of its multiple fibrous valve rows in teleosts. The relevant fossils are from the extinct pachyrhizodontid fish *Rhacolepis buccalis* (*Agassiz, 1841*), known from fossils of remarkable three-dimensional (3D) preservation (*Maisey, 1994*). The fossils were collected from the Romualdo Member of the Santana Formation, a Cretaceous Konservat Lagerstätte in the Araripe Basin in the Northeast of Brazil. A pollen and spore-based biostratigraphical analysis indicates a temporal range from 119 to 113 Ma for the strata in which the fossils are found (*de Moraes Rios-Netto et al., 2012*).

*Rhacolepis buccalis* is one of the most abundant fishes in the Santana Formation (*Maisey, 1991*) and belongs to the extinct Mesozoic clade Pachyrhizodontoidei (*Arratia, 2008*; *2010*). The relationships of Pachyrhizodontoidei among teleosts have been disputed (*Taverne, 1974*; *Taverne, 1976*; *Forey, 1977*). There is, however, mounting evidence that some features unite Pachyrhizodontoidei with Elopomorpha (*Maisey, 1991*) and, recently, Pachyrhizodontoidei, among Crossognathiformes, were placed as a group closely related to Elopomorpha in a basal position among all living teleosts (*Arratia, 2008*; *2010*). Thus, because of its basal phylogenetic position, *R. buccalis* is well suited for studies on the evolution of morphological characters in teleosts.

Here, we report complete fossil hearts from two *R. buccalis* specimens (*Figure 1*). The fossils were scanned with propagation phase contrast synchrotron radiation microtomography (PPC-SR-µ CT) at 6 µm resolution The remains of the *R. buccalis* heart are compressed along the latero-medial axis and were found in an orthotopic position, that is, posterior to gills and between the bones of the pectoral girdle (*Figure 1a–b*, *Video 1*). Their cardiac affinity is inferred on the basis of an S-shaped configuration, four chambers (conus arteriosus, ventricle, atrium and sinus venosus), typical ventricular (thick, arrowheads) and atrial (thin, arrows) muscular trabeculae (*Figure 1c–f*; *Video 2*), as well as paired Cuvier ducts that join the sinus venosus in the posterior-most region of the heart (not shown).

The outflow tract in *R. buccalis* displays a well-defined conus arteriosus encased by pericardium. Our observations indicate that the *R. buccalis* conus arteriosus is formed by a thick muscular wall, displays the morphology of a cylinder, which eventually tapers off before joining the aorta at its cranial end and is endowed with multiple valve rows (*Figure 1c–d*, *Video 2*, *Figure 2*). In the region immediately apposed to the heart, the pericardial sac assumes the shape of a pyramid wedged between posterior bilateral gill regions (*Video 2*). In one of the specimens (CNPEM 17P), the pericardial layer is easily identified near the conal myocardial wall (*Figure 2p*), while in the other specimen (CNPEM 01P), the limits between the conus arteriosus muscular wall and the pericardium are less marked (*Figure 2q*). In summary, *R. buccalis* heart is unique among teleosts in that it displays a large, dominant, conus arteriosus, rather than a predominant bulbus arteriosus in its outflow tract.

Inside the fossilized conus arteriosus it is possible to discern multiple, nearly parallel layers (*Figure 2*), which, upon 3D reconstruction, appear disposed as helicoid rings along the cranio-caudal axis of the chamber (*Figures 2b,e,h,k,n*, *Video 3*). These structures are interpreted as the fossilized remnants of the fibrous component of individual conal valves, presumably valve leaflets. For

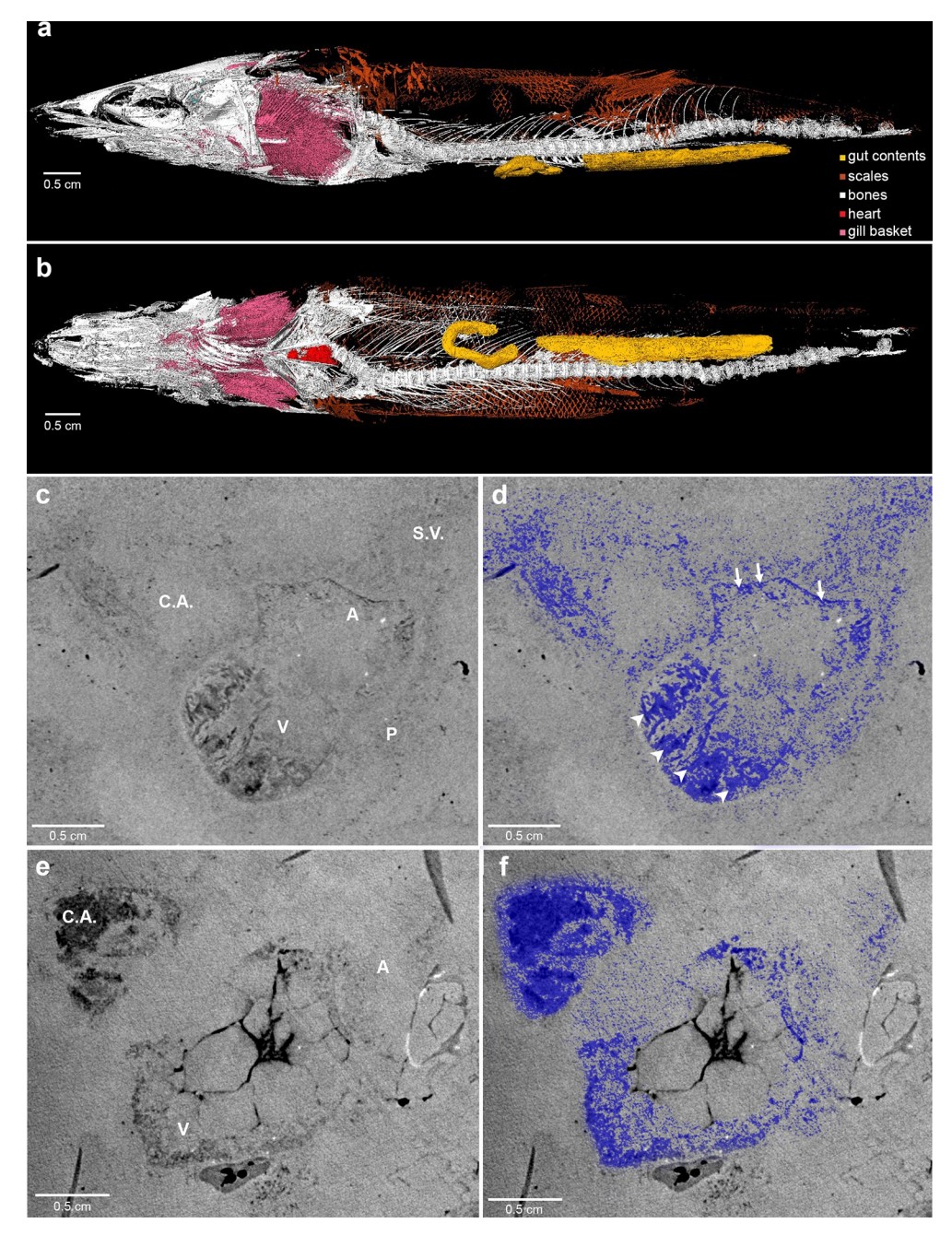

**Figure 1.** Phase contrast synchrotron micro tomography of teleost fossil hearts. (**a,b**) 3D reconstructions of specimen CNPEM 27P obtained from PPC-SR-µCT. (**a**), Left lateral view. (**b**), Ventral view. (**c,d**), (**e,f**) Sagittal sections of specimens CNPEM 01P and CNPEM 17P, respectively. Blue masks in (**d**) and (**f**) highlight fossil cardiac chambers and pericardium in the specimens CNPEM 01P and CNPEM 17P, respectively. Note that thin trabeculae are associated to the atrium (arrows) and that thick trabeculae are typical of the ventricle (arrowheads) Abbreviations: A, atrium; C.A., conus arteriosus; P, pericardium; S.V., sinus venosus; V, ventricle.

comparison, we depict the fibrous components (valve leaflets) of the two conal valves of *Megalops atlanticus* (*Figure 3*), a living basal teleost, related to *R. buccalis*.

Sagittal and coronal tomographic sections and 3D reconstructions in the two fossil specimens are consistent with the presence of at least five valve rows per conus arteriosus (*Figures 2c,f,l,o,p,q*).

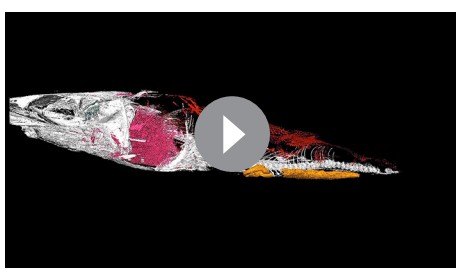

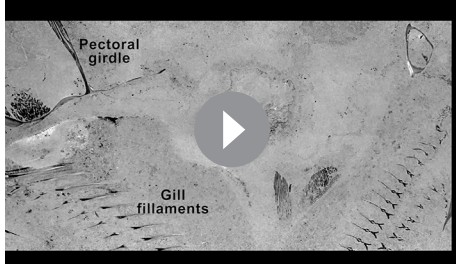

**Video 1.** 3D reconstruction of *Rhacolepis buccalis* CNPEM 27P PPC-SR-µCT. Animated rotation of the whole specimen zooming at heart position.

**Video 2.** *Rhacolepis buccalis* PPC-SR-µCT. Details of tomography at the heart region and 3D reconstruction of the conal valves.

Because of post-mortem changes, of the imperfect alignment of the conus arteriosus to the body axes, and of the semi-lunar character of conal valves (*Figure 2*), the transverse sections shown in *Figure 2* actually represent shallow oblique sections that allow the depiction of more than one valve row per transverse plane (*Figure 2b,e,h,k,n*), although it is difficult to describe with precision the exact number of valves in each valve row due to the incomplete state of preservation.

## Discussion

One important issue in the study of evolution is the idea of direction, that is, whether natural selection intrinsically favors the emergence of more complex forms or not (*McShea, 1996*). However, an unequivocal association of evolution with complexity is not a requirement of evolutionary theory (*Darwin and Wallace, 1858*). Moreover, such a view is at odds with biological evidence of frequent secondary simplifications in the evolution of microorganisms, parasites and in miniaturized/cave/fossorial fishes (*Lwoff, 1943*; *Brusca and Brusca, 2003*; *Britz et al., 2014*). Indeed, cases of morphological simplification reported in the literature most likely represent only a fraction of the examples that falsify the notion that evolution must lead to increased complexity. Many other examples of simplification are found in the evolution of animal circulatory systems (*Xavier-Neto et al., 2010*; *Brusca and Brusca, 2003*). Was this the result of traditional phyletic gradualism, or of a saltation event in the wake of large-scale gene duplication (*Amores et al., 1998*)?

The five rows of conal valves of *R. buccalis* contrast to the nine valvar rows, each containing three to six individual valves in the basal actinopterygian *Polypterus* (*Durán et al., 2014*). *Rhacolepis buccalis* valves also stand out when compared to the very limited number of conal valves in living teleosts: two valve rows in Elopomorpha (excepting *Elops*, with one) and a single valve row at the bulbo-ventricular transition in remaining teleosts. Taken together, these two characters, valvar content and overall composition of the *R. buccalis* heart (i.e. chamber vs. elastic segment), suggest that the outflow tract of this extinct fish represents an intermediate morphology between basal and higher actinopterygians, frozen in time by fossilization as an evolutionary picture taken at the Aptian/Albian boundary, 119–113 Ma (*Rios-Netto et al., 2012*) (*Figures 2* and *3*).

Valvar reduction in Actinopterygii was neither seamless, nor restricted to the teleost clade. In fact, acipenseriforms and amiiforms display independent evolutionary tendencies for conus arteriosus simplification and valve reduction when compared to *Polypterus*. Moreover, among chondrosteans, only acipenseriforms display a clear phenotype of valve reduction, while within holosteans only amiiforms show a reduced number of valves. This suggests that valve reduction happened at least three times independently in actinopterygians (*Figure 3*).

The five fossil valve rows we describe in *R. buccalis* indicate that the process of outflow tract simplification involved at least three major transitions at the base of the teleost radiation (around 284 Ma (*Betancur et al., 2013*; *Broughton et al., 2013*)): one from nine valve rows to five valve rows (e.g. from *Polypterus* to *R. buccalis*); another from five to two valve rows (e.g. from *R. buccalis* to living elopomorphs); a third to the single outflow valve row retained in all other teleosts (*Figure 3b*).

It is important to recognize that events other than simplification are also involved in the evolution of the outflow tract in vertebrates. For instance, there is evidence for increase in the number of

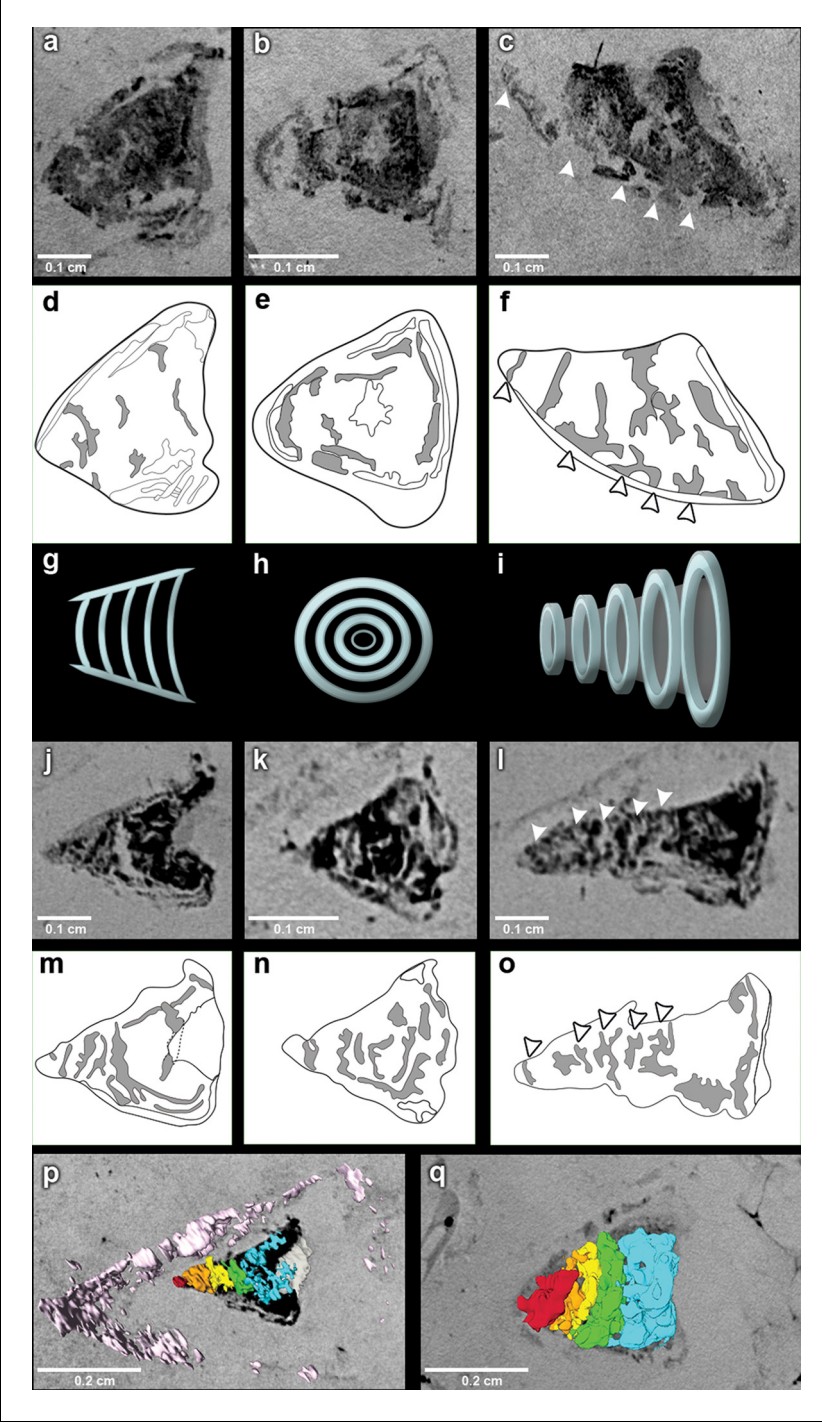

**Figure 2.** The fossil conus arteriosus of *Rhacolepis buccalis.* (a-c) Coronal, transversal and sagittal sections of the conus arteriosus of specimen CNPEM 17P taken by Phase contrast synchrotron microtomography (PPC-SR-µCT), respectively. Arrowheads in (c) indicate five conal valve rows in sagittal perspective. (d-f), Drawings of sections in (a-c) highlight conal valve rows (gray). (g-i) Didactic scheme to indicate the orientation of individual valve rows along the three orthogonal body planes (a-c) and (j-l), (j-l) Coronal, transversal and sagittal sections of the conus arteriosus of specimen CNPEM 01P taken by PPC-SR-µCT. Arrowheads in (l) indicate five individual conal valves in sagittal perspective. (m-o) Drawings of sections in (j-l) represent conal valves (gray). (p-q), 3D reconstruction and segmentation of conal valves from specimens CNPEM 01P and 17P, respectively. Note that the pericardium (pink) outlines the conus arteriosus (p). Each individual conal valve is represented by a specific spectral color.

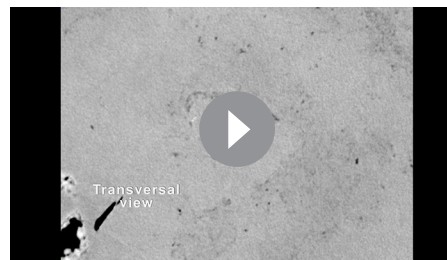

**Video 3.** *Rhacolepis buccalis* PPC-SR-μCT. Sections of conal valve.

valves occurring independently in basal Actinopterygii clades, explicitly in Polypteriformes and Lepisosteiformes, which is not illuminated by our present findings.

Currently, it is not possible to ascertain genetic correlates for the valve reduction event in *R. buccalis*. However, this does not prevent informed speculation that may set parameters for investigation in extant species with longer, or shorter, divergence times from the exuberantly valved *Polypterus*. In this sense, it is useful to observe that valve reductions in acipenseriforms and amiiforms are uncoupled from the teleost extra round of large-scale genome duplication that may, or may not, have affected *R. buccalis*. This suggests the possibility that slow, smaller scale, mutational events produced incremental phenotypic changes, which may have been gradually selected for outflow tract simplification in teleosts (*Shapiro et al., 2004*).

What developmental mechanisms could underlie the transition from conal to bulbal dominance and from valve-rich to single-valved outflow tracts? Although we deal here with outflow composition and number of valve rows as independent characters, it is possible that these traits are not completely independent, and that the relevant parameter is simply the relative extent of outflow tract occupied by the conus arteriosus and its valves, or by the bulbus arteriosus and its valveless, elastic, character (*Munoz-Chapuli et al., 1997*). Outflow tract variability among actinopterygians was modeled according to Turing's reaction-diffusion paradigm (*Munoz-Chapuli et al., 1997*). The major conclusion of this exercise was that the various rows of valves distributed across the cranio-caudal extent of the conus arteriosus and the interspersed valveless spaces can be described as an ensemble of multiple positive and negative domains of endothelial to mesenchymal transformation

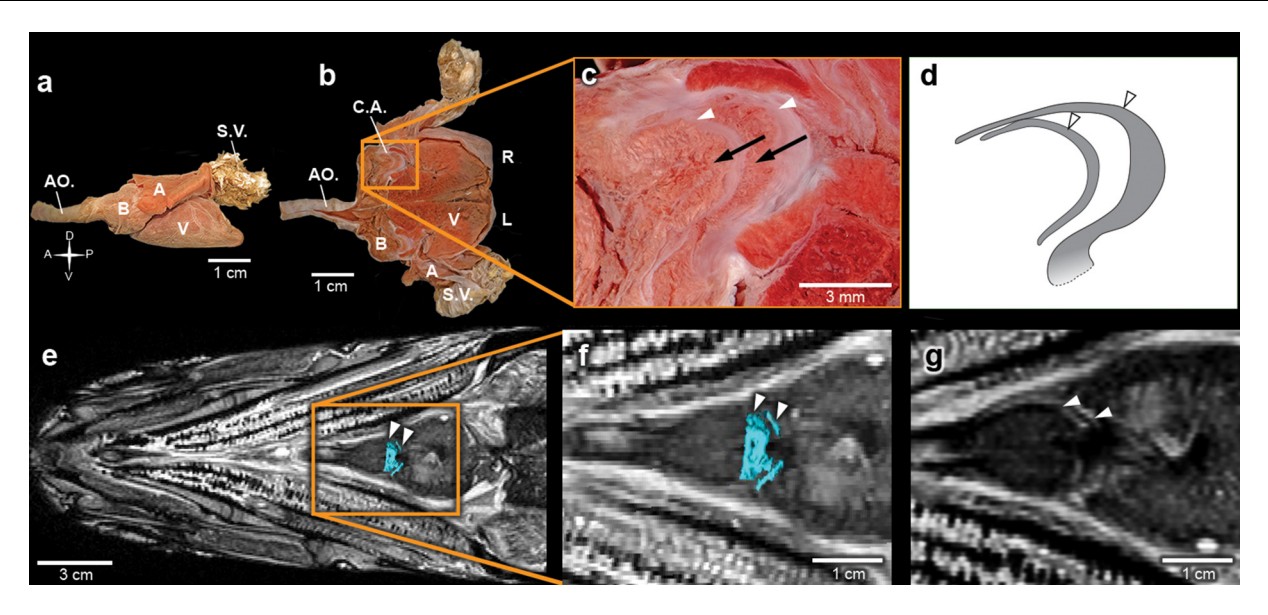

**Figure 3.** The heart of the extant elopiform *Megalops atlanticus* with a focus on its outflow tract. (**a**) Dissected heart of *M. atlanticus*. (**b**) The *M. atlanticus* heart was cut open along the sagittal plane to expose right and left components of the two conus arteriosus valves. (**c**) Magnification of the conus arteriosus in (**b**) showing valve leaflets from the two valve rows (white arrowheads) and the endocardial surface overlying conus arteriosus muscles (black arrows). (**d**) Scheme representing the right valve leaflets from the conus arteriosus of *M. atlanticus* as displayed in (**c**). (**e**) 3D reconstruction and segmentation of conal valves (blue) superimposed on a *M. atlanticus* Magnetic Resonance Imaging (MRI). (**f**) Detail of (**e**). (**g**) MRI of the *M. atlanticus* outflow tract, highlighting two conal valves (arrowheads). Abbreviations: A, atrium; AO., aorta; B, bulbus arteriosus; C.A.; conus arteriosus; L, left side; R, right side; S.V., sinus venosus; V, ventricle.

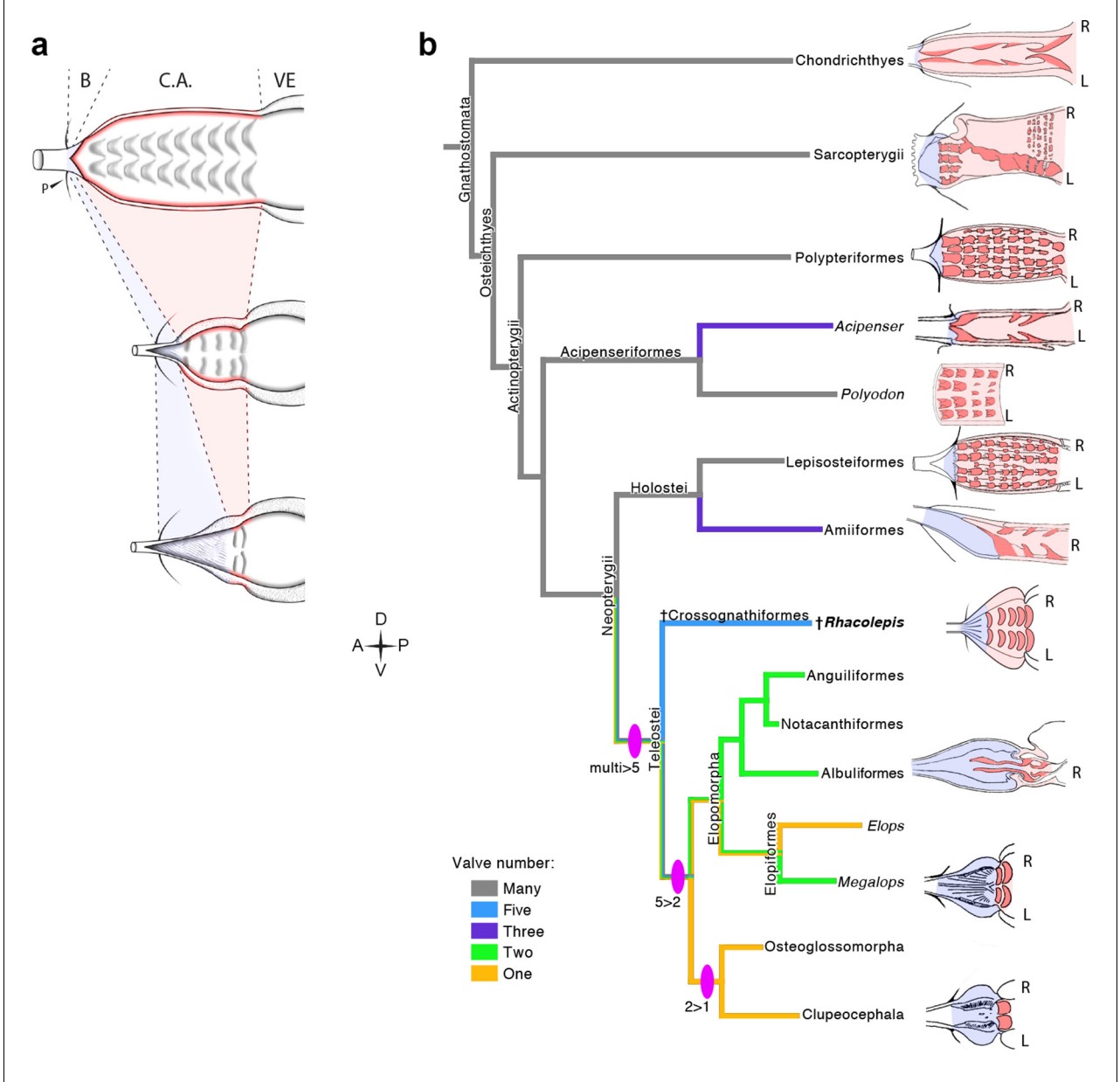

**Figure 4.** The *Rhacolepis buccalis* conus arteriosus is morphologically intermediate in actinopterygian cardiac outflow tract evolution. (a) Hypothetical transition from a character state composed by an array of multiple valve rows in the conus arteriosus of basal actinopterygians, such as Polypteriformes (top), to a derived state characterized by the dominance of the valveless bulbus arteriosus, in living teleosts (here represented by a generalized elopomorph at the bottom), through an intermediate state represented in the conus arteriosus of fossilized *R. buccalis* hearts (middle). Anterior to left. (b) Cladogram depicting phylogenetic relationships among early and derived gnathostomes and their corresponding morphologies of the cardiac outflow region. Drawings represent either the inner sides of right (R) and left (L) counterparts, or only the inner right side of the cardiac outflow tract. Drawings were modified from classic illustrations (*Parsons, 1929*; *Danforth, 1912*; *Senior, 1907*) (not to scale). Blue and pink coloring highlight, respectively, bulbus and conus arteriosus (and respective valves) in extant species. Valvar arrangement in *Rhacolepis* is suggested by data in *Figure 2*. A parsimony ancestral character state reconstruction was made for the number of conal valves, following the color code in terminals. General relationships of Teleostei were based on *Arratia, 2010*. Genera illustrating the conal condition in each Actionopterygian branches are: *Squalus* for Chondrichthyes; *Neoceratodus* for Sarcopterygii; *Polypterus* for Polypteriformes; *Lepisosteus* for Lepisosteiformes; *Amia* for Amiiformes; *Pterothrissus* for Albuliformes; *Gadus* for Clupeocephala. Abbreviations: B, bulbus; C.A., conus arteriosus; L, left side; P, pericardium; R, right side; VE, ventricle.

(*Runyan and Markwald, 1983*) set up by the interaction between diffusible molecules playing activator and inhibitor roles (*Munoz-Chapuli et al., 1997*).

The data now available suggest a case of phyletic gradualism, rather than an abrupt saltation-like event for actinopterygian outflow tract simplification. Three lines of evidence support this speculation: evidence for three independent (i.e. convergent) events of valve reduction in Actinopterygii (in acipenseriforms, amiiforms and teleosts); the three valvar simplification steps in teleosts (multiple to five, five to two, and two to one; *Figure 4*) and the inferred simplicity of developmental mechanisms capable of producing these phenotypes.

The discovery of a fossil heart in *R. buccalis* demonstrates that systematic, non-destructive approaches can be employed to study cardiovascular evolution and suggests that these sensitive techniques can be utilized not only in the context of species associated with abundant fossils, but also with rare fossils of animals at key phylogenetic positions. Regardless of these specific questions, we hope our results will open exciting new possibilities for research in cardiovascular paleontology and evolution.

## Material and methods

### Examined material

The *Rhacolepis buccalis* fossils used in this study were collected from the Romualdo Member of the Santana Formation, in the Cretaceous of Araripe Basin in the Northeast of Brazil. They are deposited in the Exceptional Preservation Collection at the Brazilian Biosciences National Laboratory (LNBio, Campinas, Brazil) and Brazilian Center for Research in Energy and Materials (CNPEM) under the following accession numbers: CNPEM 01P; CNPEM 17P; CNPEM 27P.

### Propagation phase contrast synchrotron radiation microtomography (PPC-SR-µCT)

Carbonatic nodules were scanned at the ID17 and ID19 beamlines of the European Synchrotron Radiation Facility (ESRF, Grenoble, France). For all samples, we set a propagation phase contrast microtomography protocol with a sample/detector distance of about 10 m. On ID17, we had a monochromatic beam (double-bended Laue crystals) of 150 keV. On ID19, we used a filtered pink beam (Wiggler W150 with a gap of 28 mm, filters: Al, 2.8 mm; Cu, 8 mm; W, 1 mm) with a total integrated energy of 210 keV. Two optic systems were utilized depending on the size of the nodules: a 0.5x magnification system with a FreLoN-2K camera resulting in a recorded isotropic pixel size of about 28 µm and a 0.3x magnification system with a FreLoN-2K camera resulting in a recorded isotropic pixel size of about 47 µm. The tomographies were computed based on 4998 projections over 360 degrees (pixel in horizontal x vertical: 1740x300 on ID19; 1800x130 on ID17). The exposure time per projection was 0.2 s on ID17 and 0.07 s on ID19. As the vertical field of view could not cover the full height of a nodule, multiple scans were necessary for each specimen, with a minimum overlap of 30% between each scan to correct the vertical profile of the X-ray beam. The reconstructed volumes were stitched together to visualize whole nodules and by optimizing the overall contrast (i.e. stretching the range of grey values from the 32 bit raw data into a 16 bit full range of values, avoiding too high levels of saturation).

### Segmentation

All three dimensional (3D) images of the reconstructed morphology of *R. buccalis* fossils were prepared with the AMIRA software, using TIFF images reconstructed from data obtained by propagation phase contrast synchrotron radiation microtomography scans. 3D models were built using the isosurface and segmentation features of AMIRA.

During segmentation of *R. buccalis*, we determined that its conal valves rows are continuous and follow a well-defined helicoid (clockwise) trajectory. The identities of each individual valves were assigned whenever the segmented coils reached the same relative position in the spiral (i.e. concluded a pitch).

## Acknowledgements

We are indebted to Y Petroff (LNLS, Brazil), J Roque (LNLS, Brazil), H Westfahl (LNLS, Brazil), K Franchini (LNBio, Brazil) and to P Tafforeau (ESRF, France) for continued support. Coordenadoria de Apoio ao Pessoal de Ensino Superior (CAPES, 01P-03488/2014) to LM, Fundacão de Amparo à Pesquisa do Estado de São Paulo (FAPESP, 2012/05152-0) to MC, Conselho Nacional de Desenvolvimento Científico e Tecnológico (CNPq, 481983/2013-9) to JXN.

## Additional information

### Funding

| Funder | Grant reference number | Author |
| --- | --- | --- |
| Coordenação de Aperfeiçoamento de Pessoal de Nível Superior | 01P-03488/2014 | Lara Maldanis |
| Fundação de Amparo à Pesquisa do Estado de São Paulo | 2012/05152-0 | Murilo Carvalho |
| Conselho Nacional de Desenvolvimento Científico e Tecnológico | 481983/2013-9 | José Xavier Neto |

The funders had no role in study design, data collection and interpretation, or the decision to submit the work for publication.

### Author contributions

LM, Produced acid preparations, rock sections, image segmentations and data integration, Acquisition of data, Analysis and interpretation of data, Drafting or revising the article; MC, Geology and Systematic Paleontology, Performed anatomical analyses, Edited figures and reviewed the manuscript, Acquisition of data, Analysis and interpretation of data, Drafting or revising the article; MRA, RSN, RJP, ROF, FR, SS, FAL, DG, CAP, JB, Validated initial findings and contributed to discussions and writing, Acquisition of data, Analysis and interpretation of data, Drafting or revising the article; FIF, JX-N, Fieldwork, Conception and design, Acquisition of data, Analysis and interpretation of data, Drafting or revising the article; JAFGdA, Fieldwork, Acquisition of data, Analysis and interpretation of data, Drafting or revising the article; CER, CT Scans, Acquisition of data, Analysis and interpretation of data, Drafting or revising the article; ISC, Fieldwork, Geology and Systematic Paleontology, Acquisition of data, Analysis and interpretation of data, Drafting or revising the article; MRdC, Geology and Systematic Paleontology, Discussed the results, reviewed and contributed to the text, Acquisition of data, Analysis and interpretation of data, Drafting or revising the article; VF, PPC-SR-μCT scans and reconstructions, Acquisition of data, Analysis and interpretation of data, Drafting or revising the article

### Author ORCIDs

Murilo Carvalho, http://orcid.org/0000-0001-6881-5271
Ismar S Carvalho, http://orcid.org/0000-0002-1811-0588
Carlos Alberto Perez, http://orcid.org/0000-0003-4284-3148
José Xavier-Neto, http://orcid.org/0000-0003-4648-789X

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
