## [Decision Letter]

Thank you for submitting your work entitled "Heart fossilization is possible and informs the evolution of cardiac outflow tract in vertebrates" for consideration by *eLife*. Your article has been reviewed by three peer reviewers, and the evaluation has been overseen by Diethard Tautz as the Reviewing and Senior Editor.

The following individuals involved in review of your submission have agreed to reveal their identity: John Long (reviewer 2) and John Masey (reviewer 3).

The reviewers have discussed the reviews with one another and the editor has drafted this decision to help you prepare a revised submission.

Summary:

This is an intriguing and novel line of research that goes a long way toward extending soft comparative anatomy into the fossil record, in much the way traditional paleontology has provided insights into the evolution of skeletal morphology.

Essential revisions:

Two major issues need to be addressed in a revised version. Referee 1 raises several concerns about terminology and anatomic interpretations. The current hypothesis put forward in the manuscript is not fully compatible with the known morphology of the outflow tract in extant species. This needs to be deeply revised as it is basic to advance any further evolutionary hypothesis.

The second major point concerns the limitations in the interpretation that can be inferred from a single specimen in time. It is difficult to argue for phyletic gradualism over punctuated equilibrium (Discussion, seventh paragraph) with just one data point as the rates of evolution are determined either by dating a series of fossils and their morphological changes over time (Eldridge and Gould 19742 original paper) or by using software like BEAST (e.g. see Lee et al. Science 2014, 345: 562) using 'super' character matrices incorporating autapomorphies into the mix (not done here). Whilst one can infer this is a likely evolutionary scenario, it is largely based on evidence from living actinopterygians. Accordingly, the discussion needs to be more critical with stating that this is a speculation about rates of evolution of the actinopterygian heart (rather than presenting new evidence based on a series of data, not just one point).

Reviewer 1 raised some major concerns that need to be addressed prior to publication, so their comments are appended below verbatim.

*Reviewer #1:*

The good thing about the present report is that, under certain conditions, heart tissues fossilize, can be studied, and can convey valuable information. However, the authors probably go beyond the level of the information that can be extracted from the specimens studied, use terminology that may be confusing and/or misleading, and may have some misconceptions on the evolution of the outflow tract in fishes.

First, the conus arteriosus, despite its name, is a cylindrical structure. The shape of a cylinder can only be recognized in Figure 1 and, less clearly, in Figure 2. I am willing to accept that the chamber the authors are studying is a conus arteriosus (the wedge shape is more compatible with a bulbus arteriosus), but the authors should stress more strongly not only on post-mortem changes (Results, last paragraph) but on possible deformations inherent to the fossilization process. This is especially important when the core of the study relates to identification of the valve tissue that is easily deformable and, most probably, more prone to modifications than, for instance, a thick muscle layer.

Regarding valve number, the authors identify five individual conal valves. This is quite confusing. Usual terminology describes the multiple conal valves of primitive fishes as arranged into longitudinal and transverse rows. Each transverse row has from two to six valves. So, by simple analogy, it is unclear whether each individual conal valve described in *R. buccalis* is, indeed, a valve row, i.e., a situation in which individual valves cannot be resolved within each row. More confusing, the panel corresponding to *R. buccalis* in Figure 4 shows, at least, two valves in each valve level.

Also regarding the number of valves (or valve rows), the fifth valve of *R. buccalis* is very tiny. This contrasts with the situation in multi-valved conus arteriosus, where the valves of the distalmost row are the best developed and appear to be the only ones to be functionally relevant.

In relation to the above two paragraphs, panels G-I of Figure 2 appear to be irrelevant based on both the conus shape and the number of rings. Indeed, the rings do not represent anything meaningful since several valves should be included in each row. Additionally, the valves are "semi-lunar" (Results, last paragraph), i.e., semilunar or pocket-like insertion into the conal wall.

As far as I understand it, the fish outflow tract never undertook a "transition from conus to bulbus" (Discussion, sixth paragraph). Rather, all gnathostomata show both conus arteriosus and bulbus arteriosus. In cartilaginous and basal bony fishes, the conus is morphologically predominant. In contrast, most teleosts show a morphologically predominant bulbus, the bulbus coexisting with a reduced, bearing-valves conus. All of this can be followed, for instance, in references: Duran et al., 2008; Schib et al., 2002; Icardo et al., 2002 and Grimes and Kirby, 2009. Also, see Anat. Rec. 288:900(2006); Zoology 117:370(2014).

I agree with the authors on phyletic gradualism and small-scale mutational events resulting in simplification of the outflow tract. In this regard, Figure 4 includes chondrosteans and bony fishes. If we look at the entire series (not just at the Actinopterygii), all the outflow tracts, from chondrichthyans to holosteans, show 4-6 valve rows (with a small reduction in most Acipenseriforms and a mixed situation in sarcopterygian species). What appears more remarkable is the existence of two peaks of valvar increase (Polypteriforms and Lepisosteiforms) over the mean 4-6 rows. Overt simplification is clearly seen within the teleost group, not before. This coincides with a drastic reduction in conus length.

[Editors' note: further revisions were requested prior to acceptance, as described below.]

Thank you for resubmitting your work entitled "Heart fossilization is possible and informs the evolution of cardiac outflow tract in vertebrates" for further consideration at *eLife*. Your revised article has been favorably evaluated by Diethard Tautz (Senior and Reviewing Editor) and one reviewer. The manuscript has been improved but there are some remaining issues that need to be addressed before acceptance, as outlined below:

The comments of the reviewer are provided in full. We expect that it should be straightforward to deal with them.

*Reviewer #1:*

This is a much-improved version of the original manuscript. The authors have done a good job being responsive to most of the queries. I still have a few comments:

There still is some confusion related to the use of "valves" and "valve rows". A single valve is formed by the leaflet and the supporting sinus. Several valves arrange transversely to form a valve row. Several valve rows organize along the length of the conus. In relation to this:

Introduction, third paragraph: The statement about *Polypterus* having "dozens of outflow tract valve rows" appears to be somewhat hyperbolized. A similar statement appears in the fourth paragraph of the Discussion. The publication of reference (Durán, 2014) indicates "nine rows of valves". This is correctly stated in the second paragraph of the Discussion.

"More than one valve per transverse plane": I understand that, in your specimens, findings are compatible with the existence of several valves per valve row, although the exact valve number could not be stated with any certainty. If so, this should be stated in a more precise way.

---

## [Author Response]

Essential revisions:

*Two major issues need to be addressed in a revised version. Referee 1 raises several concerns about terminology and anatomic interpretations. The current hypothesis put forward in the manuscript is not fully compatible with the known morphology of the outflow tract in extant species. This needs to be deeply revised as it is basic to advance any further evolutionary hypothesis. The second major point concerns the limitations in the interpretation that can be inferred from a single specimen in time. It is difficult to argue for phyletic gradualism over punctuated equilibrium (Discussion, seventh paragraph) with just one data point as the rates of evolution are determined either by dating a series of fossils and their morphological changes over time (Eldridge and Gould 19742 original paper) or by using software like BEAST (e.g. see Lee et al. Science 2014, 345: 562) using 'super' character matrices incorporating autapomorphies into the mix (not done here). Whilst one can infer this is a likely evolutionary scenario, it is largely based on evidence from living actinopterygians. Accordingly, the discussion needs to be more critical with stating that this is a speculation about rates of evolution of the actinopterygian heart (rather than presenting new evidence based on a series of data, not just one point).*

Reviewer 1 raised some major concerns that need to be addressed prior to publication, so their comments are appended below verbatim.

Reviewer #1:

*The good thing about the present report is that, under certain conditions, heart tissues fossilize, can be studied, and can convey valuable information. However, the authors probably go beyond the level of the information that can be extracted from the specimens studied, use terminology that may be confusing and/or misleading, and may have some misconceptions on the evolution of the outflow tract in fishes.*

We have considered reviewer 1’s comments and hope to have introduced the necessary changes to eliminate the aforementioned issues.

*First, the conus arteriosus, despite its name, is a cylindrical structure. The shape of a cylinder can only be recognized in Figure 1 and, less clearly, in Figure 2. I am willing to accept that the chamber the authors are studying is a conus arteriosus (the wedge shape is more compatible with a bulbus arteriosus), but the authors should stress more strongly not only on post-mortem changes (Results, last paragraph) but on possible deformations inherent to the fossilization process. This is especially important when the core of the study relates to identification of the valve tissue that is easily deformable and, most probably, more prone to modifications than, for instance, a thick muscle layer.*

We appreciate the reviewer’s comments on the morphology of the conus arteriosus. Accordingly, we have changed the text in which we mention the typical cone-or spade shaped morphology of the chamber. The text now reads:

“The *R. buccalis* conus arteriosus is formed by a thick muscular wall, displays the morphology of a cylinder, which eventually tapers off before joining the aorta at its cranial end, and is endowed with multiple valve rows (Figure 1_D, Video 2, Figure 2).”

*Regarding valve number, the authors identify five individual conal valves. This is quite confusing. Usual terminology describes the multiple conal valves of primitive fishes as arranged into longitudinal and transverse rows. Each transverse row has from two to six valves. So, by simple analogy, it is unclear whether each individual conal valve described in R. buccalis is, indeed, a valve row, i.e., a situation in which individual valves cannot be resolved within each row. More confusing, the panel corresponding to R. buccalis in Figure 4 shows, at least, two valves in each valve level.*

We apologize for not representing the number of valves of *R. buccalis* with enough precision. We believe we have successfully identified why reviewer 1 feels our drawings are confusing, but this has to do with the historical sources for our figures, rather than with our concepts.

In Figure 4 we adapted drawings from historical, classic, manuscripts. In those manuscripts, the outflow tract was represented in two different forms, as per original drawing: 1) Cut in two halves, with only the right half being displayed; or 2) Partially cut, open to expose all valve rows.

Apparently, reviewer 1 interpreted (not without reason) that Figure 4 displays only the inner surface of the right sagittal half (e.g. as depicted for chondrichthyans, acipenserids, amiids and albulids) in all schemes. However, as per original drawings, Figure 4 also includes partially cut, open outflow tracts that expose all valve rows from both sides (e.g. as depicted for sarcopterygians, polypterids, *Polyodon, Rhacolepis*, megalopids and clupeids).

Below we list the clades displayed in Figure 4 and the different forms of representation for each individual group.

**Clade****Method of display in drawings**ChondrichthyansCut in two halves, right half displayedSarcopterygiansPartially cut, open to expose all valve rowsSarcopterygiansPartially cut, open to expose all valve rowsPolyperidsPartially cut, open to expose all valve rowsAcipenseridsCut in two halves, right half displayedPolyodonPartially cut, open to expose all valve rowsLepisosteidsCut in two halves, right half displayedAmiidsCut in two halves, right half displayedRhacolepisPartially cut, open to expose all valve rowsAlbulidsCut in two halves, right half displayedMegalopidsPartially cut, open to expose all valve rowsClupeidsPartially cut, open to expose all valve rows

In summary, in response to reviewer 1’s concerns, and to avoid confusion, we changed Figure 3 and Figure 4 and their legends to indicate exactly which ones are the outflow tract halves depicted (right, left, or both). More specifically, in Figure 3 and Figure 4 we label the right (R) and left (L) halves of each outflow tract, as indicated in the table above. We now hope the source of misunderstanding is eliminated and thank reviewer 1 for the observations.

*Also regarding the number of valves (or valve rows), the fifth valve of R. buccalis is very tiny. This contrasts with the situation in multi-valved conus arteriosus, where the valves of the distalmost row are the best developed and appear to be the only ones to be functionally relevant.*

Unfortunately, it is not possible for us to speculate meaningfully on which individual fossil valve played the major role in the living animal *R. buccalis*. However, the reviewer points to an important question, which is the criterion we utilized to estimate the number of valves in the fossils.

During segmentation of *R. buccalis*, we determined that its conal valves rows are continuous and follow a well-defined helicoid (clockwise) trajectory. The identities of each individual valves were assigned whenever the segmented coils reached the same relative position in the spiral (i.e. concluded a pitch). This information was added to the Materials and methods section. Please note that the number of valves in the conus arteriosus of *R. buccalis* is indicated in Figure 2, in which we show quite clearly that presence of possesses at least 5 valves in specimen01P.

To clarify the issue, we inserted the criterion for determination of valve number in the Materials and methods section describing the segmentation procedure.

*In relation to the above two paragraphs, panels G-I of Figure 2 appear to be irrelevant based on both the conus shape and the number of rings. Indeed, the rings do not represent anything meaningful since several valves should be included in each row. Additionally, the valves are "semi-lunar" (Results, last paragraph), i.e., semilunar or pocket-like insertion into the conal wall.*

Panels G-I in Figure 2 are schematic drawings depicted for didactic purposes, mostly to facilitate the comprehension of what is the actual 3D distribution of a valve row to readers who are not necessarily specialists in the field. We believe those diagrams make it easier for the non-specialist reader to evaluate the fossil data in Figure 2 and, as such, we would like to keep them, if possible.

Nonetheless, we sought to improve the clarity of our schematic drawings by stating in Figure 2 legend that the rings represent valve rows.

*As far as I understand it, the fish outflow tract never undertook a "transition from conus to bulbus" (Discussion, sixth paragraph). Rather, all gnathostomata show both conus arteriosus and bulbus arteriosus. In cartilaginous and basal bony fishes, the conus is morphologically predominant. In contrast, most teleosts show a morphologically predominant bulbus, the bulbus coexisting with a reduced, bearing-valves conus. All of this can be followed, for instance, in references: Duran et al., 2008; Schib et al., 2002; Icardo et al., 2002 and Grimes and Kirby, 2009. Also, see Anat. Rec. 288:900(2006); Zoology 117:370(2014).*

We thank the reviewer for flagging this inconsistency. Indeed, as we made clear in the Abstract, we fully understand that what happened in teleosts was a shift in the relative importance of the conus arteriosus to the bulbous arteriosus (e.g. please read Abstract in the original submission), not the conversion from conus to bulbus, as was implied in the aforementioned sentence.

To correct the information conveyed we rephrased the sentence, which now reads:

“What developmental mechanisms could underlie the transition from conal to bulbal dominance and from valve-rich to single-valved outflow tracts?”

*I agree with the authors on phyletic gradualism and small-scale mutational events resulting in simplification of the outflow tract. In this regard, Figure 4 includes chondrosteans and bony fishes. If we look at the entire series (not just at the Actinopterygii), all the outflow tracts, from chondrichthyans to holosteans, show 4-6 valve rows (with a small reduction in most Acipenseriforms and a mixed situation in sarcopterygian species).*

It is very important to us that reviewer 1 agrees with our speculation that phyletic gradualism is the more likely evolutionary scenario for outflow tract evolution in teleosts.

We based our conjecture on three taxa (polypterids, *R. buccalis* and clupeids) rather than in just one taxa (i.e. *R. buccalis*). We acknowledge that there is not enough data to prove that the phyletic hypothesis. However, the case for the competing hypothesis (i.e. a saltation event) is even less likely because, collectively, data from extant species seem to contradict it in at least three instances, as we refer in the text. This is because the process of outflow tract simplification involved at least three major transitions. One transition at the base of the teleost radiation (around 284 Ma (Betancur et al., 2013, Broughton et al., 2013)) from dozens of valves to five valves (e.g. from *Polypterus*, to *R. buccalis*); another transition from five to two valves (e.g. from *R. buccalis* to living elopomorphs – except *Elops*); and yet another transition to the single outflow valve retained in all other teleosts (Figure 3). It seems to us that these three transitions alluded above fit much better into the model of phyletic gradualism, than into the alternative hypothesis of saltation.

At any rate, according to the editor’s suggestions, we toned down the nature of the discussion to reflect the speculative nature of the argument. We hope our first description of fossilized conal valves will stimulate researchers to examine high quality specimens with appropriate techniques and contribute to help settle this question (please see Discussion, eighth paragraph).

What appears more remarkable is the existence of two peaks of valvar increase (Polypteriforms and Lepisosteiforms) over the mean 4-6 rows. Overt simplification is clearly seen within the teleost group, not before. This coincides with a drastic reduction in conus length.

We also agree that if one wants to tell the complete story of valve evolution in vertebrates, one must acknowledge the valvar increases that happened in polypteriforms and lepisosteiforms. However, because our findings are limited to teleosts, and are not pertinent to the evolutionary transitions between chondrichthyans and bony fishes, we did not comment on that in the original version.

We now inserted a phrase in the Discussion to comment briefly on valve gain associated with the evolution of polypteriforms and lepisosteiforms (Discussion, fifth paragraph).

[Editors' note: further revisions were requested prior to acceptance, as described below.]

*Reviewer #1: This is a much improved version of the original manuscript. The authors have done a good job being responsive to most of the queries. I still have a few comments:*

*There still is some confusion related to the use of "valves" and "valve rows". A single valve is formed by the leaflet and the supporting sinus. Several valves arrange transversely to form a valve row. Several valve rows organize along the length of the conus. In relation to this:*

*Introduction, third paragraph: The statement about Polypterus having "dozens of outflow tract valve rows" appears to be somewhat hyperbolized.*

Thank you for pointing out the inconsistency (we mistakenly kept valve rows, instead of valves). We have now changed the text to reflect the correct view. The text now reads:

“The transition from a heart packed with dozens of outflow tract *valves* in basal actinopterygians…”

A similar statement appears in the fourth paragraph of the Discussion. The publication of reference (Durán, 2014) indicates "nine rows of valves". This is correctly stated in the second paragraph of the Discussion.

Thank you again. We corrected the text and it now reads:

“…One from *nine valve rows* to five *valve rows* (e.g.…).”

*"More than one valve per transverse plane": I understand that, in your specimens, findings are compatible with the existence of several valves per valve row, although the exact valve number could not be stated with any certainty. If so, this should be stated in a more precise way.*

We have incorporated the suggestion into the text, which now reads:

“Because of post-mortem changes, of the imperfect alignment of the conus arteriosus to the body axes, and of the semi-lunar character of conal valves (Figure 2), the transverse sections shown in Figure 2 actually represent shallow oblique sections that allow the depiction of more than one *valve row* per transverse plane (Figure 2), *although it is difficult to describe with precision the exact number of valves in each valve row due to the incomplete state of preservation*.”